# Women's use of intimate apparel as subtle sexual signals in committed, heterosexual relationships

**Lyndsey K. Craig** *, **Peter B. Gray**

Department of Anthropology, University of Nevada, Las Vegas, NV, United States of America

* craigl1@unlv.nevada.edu

## Abstract

Current literature on women's sexual signaling focuses on modes of attracting potential, new sexual partners, but says little about women's subtle sexual signals in committed, romantic relationships. Subtle sexual signals are inherently private and are only visible to the intended audience; a woman might use these signals to elicit or accept a sexual response from her partner or to increase her overall attractiveness, or attractivity. In this study, we sought to identify women's use of intimate apparel as a proceptive or receptive behavior as well as the effects of relative mate value, relationship commitment, relationship satisfaction, and sexual functioning. A total of $N = 353$ women in the United States aged 25–45 who were in committed, heterosexual relationships completed the survey; 88.7% of the sample indicated wearing or having worn sexy underwear. Results indicate that women report wearing sexier underwear the day taking the survey if they anticipate sexual activity that same day. However, during the most recent sexual activity, women did not report wearing sexier underwear if they initiated (proceptive) that activity. While relative mate value was not directly related to sexiness of intimate apparel, women who report higher mate value tend to wear sexier underwear. Women's use of intimate apparel might be viewed as a method of increasing attractivity and underlying receptivity to aid relationship maintenance, though caveats regarding measures and alternative interpretations are also discussed. Findings suggest that these women use intimate apparel to feel sexy, desired, aroused, and to prepare for sex with their partners. This study is the first to examine intimate apparel in relationships and as a subtle sexual signal of proceptivity and receptivity.

## Introduction

Research on human female sexual signaling indicates many women use appearance modification strategies–for example, make-up, clothing, and cosmetic surgery–to increase their attractiveness as a means of enticing potential sexual partners (e.g., [1–3]). But these appearance enhancement strategies are broadcast signals: behaviors viewable to the general public which might be perceived as insincere in the commitment to the relationship [4]. Conversely, subtle signals are behaviors that are only visible to the person or persons the signals are directed

**Data Availability Statement:** All data files are available from the figshare database (DOI: 10.6084/m9.figshare.10316015).

**Funding:** Funding for participant recruitment was provided by the UNLV Angela Peterson Scholarship

to author LKC. In addition, UNLV funded the journal publication costs. The funder had no role in study design, data collection and analysis, decision to publish, or preparation of the manuscript.

**Competing interests:** The authors have declared that no competing interests exist.

toward. In interpersonal relationships, these behaviors tend to demonstrate more assurance and commitment to the partnership due to their inherent privacy [4, 5]. In this study, we propose a potential subtle sexual signal within romantic relationships: the use of intimate apparel, i.e., sexy underwear. Additionally, we examine women's use of intimate apparel within the context of relative mate value.

## Women's sexual signaling

In many non-human primate species, females undergo distinct physical changes during estrus which serve as approximate signals of their fertile state, such as exaggerated sexual swelling [6]. Behavioral stimuli can also be means for signaling fertility in non-human primate species [6]. Humans face unique challenges to signaling fertility due to the lack of obvious, physical cues and the regular use of clothing or other materials that hide overt evaluation of sexual status. Behavioral stimuli which females use to initiate sexual contact with male conspecifics, enticing or inviting potential mates, are labeled as proceptivity [7–10]. Alternatively, receptivity is female behavior that is conducive to penile-vaginal penetration which can be quantified by female acceptance, refusal, or termination of male mounting attempts [7, 9]. Both categories of behavior intend to elicit a sexual response but differ by who initiates. For heterosexual humans, a woman's behavior which initiates sexual contact with a man is *proceptive* whereas a woman's response to expressed sexual interest from a man is *receptive*. This paper focuses on women's proceptive behaviors in romantic relationships.

The literature on women's sexual signaling is robust but mixed, especially regarding hormonal influences across the ovulatory cycle. Research on non-behavioral stimuli indicates that men are more attracted to women's smells and appearances during their fertile state [11, 12]. Alternatively, studies of behavioral stimuli include women's use of revealing clothing, makeup, and even the color red during the fertile period of their cycles [13–15], though results are mixed and controversial. Many women undergo surgeries to enhance lips, increase breast size, make skin more taught, and ultimately fit more culturally normative expectations of attractiveness; most cosmetic surgeries worldwide are performed on women [16].

Hill and Durante [17] found that women who are actively seeking a mate are more likely to use risky strategies to enhance their appearance, such as taking diet pills or tanning; and women who are more interested in receiving cosmetic surgery tend to prefer men of higher status and attractiveness [18]. In most societies, the more attractive women have full lips, relatively large breasts, clear skin, lustrous hair, and lower waist-to-hip ratios (see [19–21]). These attributes are thought to be cues of female youth, reflecting reproductive value. These enhancement behaviors are indicative of adaptive functions of female beauty and reproduction [21] and can increase women's overall attractiveness regardless of their reproductive state. But research on women's sexual signaling largely fail to capture the woman's sexual intent regarding those behaviors, especially within committed, romantic relationships.

## Relationship maintenance and mate value

Relationship maintenance strategies include positivity, openness, social network, and task sharing [5]. Originally defined by Canary and Stafford [22], positivity is acting nice and courteous; openness involves speaking openly about thoughts and feelings; social network is spending time with mutual friends; and task sharing involves sharing in household responsibilities. Others indicate that sharing activities with a romantic partner is beneficial for relationship maintenance [23], and sexual satisfaction in relationships is associated with relationship satisfaction and commitment [24, 25]. Indeed, all these strategies are associated with greater

relationship satisfaction and commitment [5, 22, 26]. However, mate value discrepancy is arguably a determining factor for whether a person uses mate retention strategies.

Individuals who perceive their mate as having a higher mate value than their own are more likely to forgive their partner's transgressions [27] and tend to use more frequent mate retention behaviors when there is a risk of their partner's infidelity [28]. Also, both men and women tend to use more vigilant and sometimes violent behaviors when their partner's perceived attractiveness is greater than their own [29]. Conroy-Beam, Goetz, and Buss [30] found that mate value discrepancy (MVD) was predictive of individuals' relationship satisfaction and trust in their partners: an individual partnered with someone of greater mate value than themselves was less trusting of that partner. Studies on MVD, however, are limited in part due to complications with operationalizing the difference between one partner's mate value and the other's mate value. Another challenge with capturing MVD is the inherent subjectivity of the concept. A person's mate value is entirely dependent on self-reports and is left to interpretation; an evolutionary biologist might argue that mate value is equivalent to that person's reproductive fitness or contribution of resources (e.g., [21]), whereas a social scientist might argue that mate value is related to a person's ability to communicate or express empathy. For the purposes of this study, however, the precise definition of mate value is irrelevant. What matters in this context is the person's perceived mate value of herself and her partner. If a person perceives her own mate value as less than her partner's, the precise reason does not matter, but rather the effect of her perception within the relationship. Presumably, her individual ideal for what is desirable in a mate will be consistent with the judgment placed on herself and her partner. Because evidence suggests mate value plays a role in women's mate retention strategies, then mate value might also have an impact on women's use of intimate apparel.

## Women's use of intimate apparel

Research on underwear has been undertaken from a variety of approaches: from the development of new underwear technology [31, 32] to marketing effectiveness and strategies [33, 34] to underwear fetishism [35]. The most robust literature is on fabric thermoregulation [36–38] and men's boxers versus briefs: i.e., tightness and sperm quality [39, 40]. Although a more than 30 billion U.S. dollar market worldwide [41], literature on women's underwear is limited, focusing on consumerism and postfeminist critiques. And no research examines how women use underwear in their romantic relationships.

Anecdotally, women tend to have preferences for their everyday underwear depending on their external wear, such as jeans, slacks, skirts or dresses and tight, loose or low-cut tops. For example, women might wear a seamless or G-string panty with a tight skirt or pair of slacks. Many women also have specialized categories for their underwear: e.g., "period panties" and "laundry-day bra." Interview data indicate women also wear certain types of underwear because they simply make them feel comfortable or sexy. Jantzen's [42] interviews with 22 women showed that many of the participants had certain underwear for when they were ill, menstruating, or doing sports; one respondent called them "amateur briefs." The women in this study indicated that the "competent" woman dresses in presentable underwear for each type of day; whether for a day in or a party, the woman should be prepared. As Jantzen [42] states, "Underwear is thus. . .a means to manipulate intra-psychological moods and generate sensations for the intimate self." These data suggest that women use certain types of underwear in preparation for their daily activities.

Sexy underwear, often broadly termed *lingerie* in the United States, can be considered distinct from daily underwear, but lingerie is an umbrella term for all types of women's underwear, not only sexy types. Therefore, for the purposes of this study, we use the terms *intimate*

*apparel* and *sexy underwear* as any underwear women intentionally use to increase their attractivity. Because women's preferences vary, intimate apparel can range from a combination of boy shorts and bralettes to crotch-less panties and leather corsets. Women aged 25–44 tend to consider purchasing lingerie and intimate apparel as a "treat" as well as a necessity [43]. While some women might wear intimate apparel to "feel good" about themselves, they more likely choose intimate apparel that increases their own arousal while also signaling sexual proceptivity and receptivity to their male partner.

## Predictions

In this study, we hypothesized that women use intimate apparel in relationships as a form of subtle sexual signaling for proceptivity. To test this, we examined women's use of intimate apparel in their current relationships across three distinct contexts: use at the time of taking the survey, use during their most recent sexual encounter with their partner, and use in an imagined, ideal sexual encounter with their partner. We then tested women's use of intimate apparel in relation to their relative mate value and with potential covariates: relationship satisfaction and commitment, sexual functioning, women's age, parental status and age of children. We predicted the following:

1. Women who intend to be sexually active later in the day will report higher sexiness of underwear on that day than women who do not intend to be sexually active.

2. Women who initiated their most recent sexual encounter with their partner will report higher sexiness of underwear for that encounter than women who did not initiate.

3. Women who report a lower relative mate value than their partner's will report higher sexiness of underwear compared to women who report the same or higher relative mate value than their partner across all three contexts:

   a. day of taking the survey

   b. most recent sexual encounter with their partner

   c. an ideal sexual encounter with their partner

## Methods

### Power analysis

An *a priori* power analysis indicated that for a power of .80 with an alpha of .05 and a moderate effect size, this study required a minimum sample size of $n = 250$ to a maximum of $n = 500$.

### Procedure

This study received exempt status from the UNLV Social/Behavioral IRB and was conducted via the online survey system Qualtrics. Links to the survey were distributed across social media sites (Facebook, Twitter, and Reddit) as well as through e-mails and flyer hand-outs; completed responses took about 15–20 minutes. The online survey began with a University of Nevada, Las Vegas standard informed consent form and four questions intended to exclude women who: were not between the ages of 25–45; were not in a committed romantic relationship; were in a long-distance relationship; or were pregnant or nursing. Participants who responded negatively to all the four were then asked a series of demographic questions designed to describe their current living arrangements, income, and romantic relationship. Next, participants answered questions about their use of intimate apparel, termed *sexy*

*underclothes* in the survey, followed by a scale about relationship satisfaction and commitment, their perceived MVD, and their sexual functioning. Participants then answered a series of questions about their current sexiness of underwear and expectancy for a sexual encounter with their current partner followed by a series of questions designed to understand their use of intimate apparel in relation to their most recent sexual encounter and an ideal sexual encounter with their current partner.

## Participants

Eight-two out of 183 respondents completed the online survey via a Qualtrics link and received an opportunity to enter a raffle to win one of three Amazon gift cards. Qualtrics Research Services then collected 268 responses for a fee of $1,500, totaling $N = 353$ completed responses. Participant ages ranged from 25 to 45 with a $M = 34$ and $SD = 6$ years. Of the 353 women who completed this survey, 80.5% reported being white, 6.8% reported being African American or black, 4.8% reported being Asian, and 7.9% reported other ethnicities.

## Measures

### Demographics

Participants were asked a series of background information about their age, their partner's age, ethnicity, education, use of hormonal contraceptives, household size, parental status and age of children, income contribution of self and partner, and whether they are living with their partner.

### Attitudes toward intimate apparel

Participants indicated if they ever wear *sexy underclothes*. We designed a 12-item questionnaire intended to assess the participants' reasons for wearing intimate apparel. The items were phrased as statements, and participants responded on a 7-point Likert scale (1 = *totally agree*; 7 = *totally disagree*). The items were based on prior research [32, 42, 44–47] on women's perceptions and attitudes toward intimate apparel (see Results for a complete list of statements).

### Use of intimate apparel

For each experimental context, participants responded to a one-item question about the sexiness of their underclothes on a continuous scale from 1–10: 1 being the least sexy; 10 being the most sexy.

### Mate value discrepancy

Mate value discrepancy (MVD) was measured using a modified version of the Mate Value Scale (MVS) [48]. The MVS is a 4-item questionnaire about a person's desirability as a partner with good reliability (Cronbach's alpha = .85) and validity. For the modification, we doubled the items and changed the second to "your partner" instead of "your." Each item used a 7-point Likert scale that differs for each question (e.g., Item 1: 1 = *extremely undesirable*, 7 = *extremely desirable*; Item 4: 1 = *very bad catch*, 7 = *very good catch*). Discrepancy was measured by subtracting the total value of items 1–4 from the total value of items 5–8. A negative score indicates the female's mate value is less than the male's mate value. A positive score indicates the female's mate value is greater than the male's mate value.

## Descriptive relationship measures

Participants were asked about their relationship satisfaction with, their commitment to, and their sexual functioning with their current romantic partner to provide a richer profile of the participants' relationship.

*Relationship satisfaction.* Relationship satisfaction was measured using the Kansas Marital Satisfaction Scale (KMSS) which is a 3-item questionnaire with a 7-point Likert scale (1 = *extremely dissatisfied*; 7 = *extremely satisfied*) and strong test-retest reliability and validity [49]. The items were modified to relate to couples who are not married but are in a committed relationship.

*Commitment.* Commitment was measuring using the Lund [50] commitment scale with good reliability (Cronbach's alpha = .82) and validity. The 9-item questionnaire uses a 7-point Likert scale that differs for each item (e.g., Item 1: "How likely is it that your relationship will be permanent?" 1 = *extremely likely*, 7 = *extremely unlikely*; Item 7: "How obligated do you feel to continue this relationship?" 1 = *extremely obligated*; 7 = *not at all obligated*). Items 2 and 6 were reverse scored.

*Sexual functioning.* Sexual function in the participants' current romantic relationship was measuring using the Female Sexual Function Index (FSFI) [51, 52]. We only used the three relevant portions of the FSFI to reduce survey length: desire (items 1 and 2; Cronbach's alpha = .92), arousal (items 3–6; Cronbach's alpha = .95) and satisfaction (items 14–16; Cronbach's alpha = .89). The standard FSFI measures sexual functioning within the last four weeks.

## Experimental contexts

Each participant responded to questions about their use of intimate apparel in the following three contexts:

*Context one.* This portion of the survey was designed to assess the participants' status of intimate apparel on the day they took the survey. These questions provide semi-random sampling of daily intimate apparel between subjects. Participants were asked if they were currently wearing or planned to wear *sexy underclothes* on the day of taking the survey and if they expected to be sexually active with their partner that day as well as rating their level of *sexy underclothes*.

*Context two.* This portion of the survey was designed to assess the participants' use of intimate apparel as a form of proceptivity during the most recent sexual encounter with their romantic partner. These questions provide an estimate for how the participants usually experience sex with their current partner. Participants answered questions about which person initiated the sexual encounter and rated their level of *sexy underclothes*. Finally, participants completed a modified version of the Female Sexual Function Index (FSFI), excluding item 6 as it could not be estimated for the most recent sexual encounter. We also removed all scores of 0 from the seven items, as we asked about sexual activity.

*Context three.* This portion of the survey was designed to assess the participants' use of intimate apparel during an ideal sexual encounter with their romantic partner. Participants read a small, hypothetical vignette with their partner initiating sexual contact and responded about their ideal level of *sexy underclothes* during the encounter.

## Statistical analysis

All data analyses were conducted using the statistical program IBM SPSS version 25.0 [53]. First, we performed descriptive, frequency, and correlational analyses on all variables to provide a demographic and relationship profile of the participants. We used an independent samples *t*-test to test the relationship of sexual intent and initiation with the use of intimate

apparel; participant responses for intent and initiation were measured as independent, dichotomous values. Although responses to the intimate apparel measure were non-normally distributed, the sample sizes are large enough to provide robust results [54]. Finally, to test our predictions we used a linear regression analysis to test for interaction effects between MVD and intent with use of intimate apparel; a linear regression analysis was also used to test for interaction effects between MVD and initiation with the used of intimate apparel.

## Results

### Descriptive analyses

Frequency data for participant demographics are listed in Table 1. Table 2 provides means and standard deviations for the measures of participant age, partner age, income contribution, relationship satisfaction, commitment, mate value, and sexual desire, arousal, and satisfaction. Finally, we provide a Pearson correlation analysis of these relationship variables in Table 3 and a Pearson correlation analysis of intimate apparel across the three contexts in Table 4.

### Intimate apparel

Of the 353 women, 88.7% ($n$ = 313) indicated wearing or having worn intimate apparel. The primary reasons for wearing intimate apparel were *to feel sexy* ($M$ = 5.65; $SD$ = 1.32), *on special occasions* ($M$ = 5.41; $SD$ = 1.39), *to feel feminine* ($M$ = 5.28; $SD$ = 1.52), *as a gift for my partner* ($M$ = 5.24; $SD$ = 1.71), and *when I expect to get intimate later that day* ($M$ = 5.20; $SD$ = 1.57). Table 5 shows a complete list of the reported reasons participants use intimate apparel.

### Prediction one

We predicted that women who intended to be sexually active later in the day would report higher sexiness of underwear. Overall, participants rated the sexiness of their current underclothes ($M$ = 4.59; $SD$ = 2.64). An independent samples *t*-test showed significant differences in sexiness of underwear between intent ($M$ = 5.68; $SD$ = 2.72) and no intent ($M$ = 3.87; $SD$ = 2.26) for sexual activity ($t_{153.09}$ = 4.48; $p$ < .001; Fig 1), providing support for prediction one. These results suggest women might wear sexier underwear when they plan on being sexually active later in the day.

### Prediction two

We predicted that women who initiated their most recent sexual encounter with their partner would report higher sexiness of underwear for that encounter than women who did not initiate. Overall, participants rated the sexiness of their underwear ($M$ = 4.97; $SD$ = 2.74) during their most recent sexual encounter with their partner. An independent samples *t*-test showed no significant difference in intimate apparel with who initiated the most recent sexual encounter (Fig 2); thus, prediction two was not supported: use of intimate apparel does not seem to be a proceptive behavior in this sample, at least based on this assessment.

To gain insight on proceptivity, we conducted an exploratory analysis with other possible confounding variables: relationship commitment and satisfaction, hormonal contraceptives, parental status, age of children, age of participant, age difference between participant and partner, income, and employment status. However, we found no significance in any of the variables regarding initiation. These results suggest intimate apparel is not a form of proceptivity or receptivity in this sample. Alternatively, these results might suggest the item used in this study did not accurately measure initiation, which was one-item asking, "Did you or your partner initiate your most recent sexual encounter?" without a clear definition for *initiate*.

**Table 1. Frequency data for participant demographics.**

| Demographic Variable | Freq. | % |
|---|---:|---:|
| **Ethnicity** | | |
| African American or Black | 24 | 6.8 |
| American Indian or Alaskan Native | 4 | 1.1 |
| Asian | 17 | 4.8 |
| White | 284 | 80.5 |
| Other | 24 | 6.8 |
| **Education** | | |
| Less than HS degree | 16 | 4.5 |
| HS degree / equivalent (e.g., GED) | 90 | 25.5 |
| Some college; no degree | 88 | 24.9 |
| Associate's degree | 39 | 11 |
| Bachelor's degree | 69 | 19.5 |
| Graduate degree | 50 | 14.2 |
| **Household size** | | |
| One (self) | 9 | 2.5 |
| Two | 113 | 32 |
| Three | 78 | 22.1 |
| Four | 82 | 23.2 |
| Five or more | 71 | 20.1 |
| **Participant employment status** | | |
| Employed up to 35 hrs/wk | 70 | 19.8 |
| Employed 36+ hrs/wk | 141 | 39.9 |
| Unemployed | 110 | 31.2 |
| Unable to work | 32 | 9.1 |
| **Household income** | | |
| $0–29,999 | 74 | 21 |
| $30,000–49,999 | 93 | 26.3 |
| $50,000–79,999 | 84 | 23.8 |
| $80,000–129,999 | 63 | 17.9 |
| $130,000+ | 39 | 11 |
| **On hormonal contraceptives** | 88 | 24.9 |
| **Married and cohabiting** | 222 | 62.9 |
| **Unmarried and cohabiting** | 105 | 29.7 |
| **Living separately** | 24 | 6.8 |
| **Participants with children** | 233 | 66 |

All variables are presented to provide an enriched profile of the participants.

## Prediction three

We predicted that women with a relatively lower mate value than their partner would report higher sexiness of underclothes across all three contexts.

**Prediction 3a.** For context one, a linear regression indicated a main effect of intent for sexual activity on sexiness of underclothes ($t$ = -6.32; $p < .001$) and a main effect of MVD on sexiness of underclothes ($t$ = 3.59; $p < .001$); however, we found no interaction effect between intent and MVD on sexiness of underclothes. In contrast to prediction 3a, the directionality of the MVD $t$-value indicates that women who have similar or higher relative mate value to their partners reported wearing sexier underwear.

**Table 2. Means and standard deviations for demographic and relationship variables.**

|  | n | M | SD | Min | Max |
|---|---|---|---|---|---|
| Age Discrepancy[1] | 349 | 2.56 | 5.53 | -21 | 28 |
| Participant Age | 352 | 34.57 | 6.05 | 25 | 45 |
| Partner's Age | 350 | 37.08 | 7.72 | 23 | 59 |
| Partner's Income Contribution[2] | 353 | 61.46 | 29.17 | 0 | 100 |
| Mate Value Discrepancy[3] | 351 | -.46 | 1.12 | -5.75 | 3 |
| Mate Value (Self) | 352 | 5.07 | 1.16 | 1 | 7 |
| Mate Value (Partner) | 351 | 5.53 | 1.13 | 1 | 7 |
| Relationship Satisfaction | 352 | 5.81 | 1.35 | 1 | 7 |
| Commitment | 350 | 5.21 | .71 | 1.78 | 7 |
| Sexual Desire[4] | 350 | 3.28 | 1.07 | 1 | 5 |
| Sexual Arousal[4] | 350 | 4.04 | 1.11 | 1 | 6 |
| Sexual Satisfaction[4] | 350 | 4.12 | 1.24 | 1 | 6 |

All variables are presented to provide an enriched profile of the participants.

[1]Variable represents the partner's age minus the participant's age. A positive score indicates the partner is older than the participant.

[2]Variable was measured in percent (%) of total household income.

[3]Variable represents the partner's mate value minus the participant's mate value. A negative score indicates the participant's mate value is less than their partner's mate value.

[4]Sexual desire, arousal, and satisfaction are individual domains of the complete FSFI measure and refer to the past four weeks.

**Prediction 3b.** For context two, a linear regression analysis showed a main effect of MVD on sexiness of underclothes ($t = 3.63$; $p < .001$). Additionally, we found no interaction effect between initiation and MVD on sexiness of underclothes. Again, the $t$-value direction shows no support for prediction 3b: women who have similar or higher relative mate value to their partners reported wearing sexier underclothes.

**Prediction 3c.** For context three, a linear regression analysis showed no effect of MVD on sexiness of underclothes during an ideal sexual encounter. Prediction 3c was not supported.

## Exploratory analysis of MVD

To better understand the role of mate value with intimate apparel, we performed an exploratory analysis on independent self and partner mate values in relation to intimate apparel

**Table 3. Pearson correlations for relationship variables.**

| Variables | 1 | 2 | 3 | 4 | 5 | 6 | 7 | 8 |
|---|---|---|---|---|---|---|---|---|
| 1. Mate Value Discrepancy[1] | – | | | | | | | |
| 2. Age Discrepancy | .071 | – | | | | | | |
| 3. Income Discrepancy | -.057 | .030 | – | | | | | |
| 4. Relationship Satisfaction | **-.202**** | .016 | **.159**** | – | | | | |
| 5. Commitment | **-.113*** | .005 | .917 | **.387**** | – | | | |
| 6. Sexual Desire[2] | .100 | .076 | .044 | **.340**** | .097 | – | | |
| 7. Sexual Arousal[2] | .028 | .025 | **.105*** | **.298**** | .079 | **.372**** | – | |
| 8. Sexual Satisfaction[2] | -.056 | -.008 | .087 | **.513**** | **.182**** | **.266**** | **.707**** | – |

*$p < .05$

**$p < .01$

[1]Negative scores indicate participant mate value was *less than* their partner's mate value.

[2]Sexual desire, arousal, and satisfaction are individual domains of the FSFI measure and refer to the *past four weeks*.

**Table 4. Pearson correlations for intimate apparel.**

| Variables | 1 | 2 | 3 | 4 | 5 | 6 | 7 |
|---|---|---|---|---|---|---|---|
| 1. *Context One*: IA Sexiness | – | | | | | | |
| 2. *Context Two*: IA Sexiness | .613** | – | | | | | |
| 3. *Context Three*: IA Sexiness | .391** | .579** | – | | | | |
| 4. Mate Value Discrepancy[1] | .215** | .208** | .038 | – | | | |
| 5. Sexual Desire[2] | .153** | .271** | .230** | .031 | – | | |
| 6. Sexual Arousal[2] | .194** | .291** | .151** | .018 | .850** | – | |
| 7. Sexual Satisfaction[2] | .158** | .204** | N/A[3] | -.104 | .647** | .750** | – |

*$p < .05$

**$p < .01$; Context One refers to participants' intimate apparel at the time of taking the survey; Context Two refers to participants' intimate apparel during their most recent sexual encounter.

[1]Positive scores indicate participant mate value was *greater than* their partner's mate value.

[2]Sexual desire, arousal, and satisfaction are individual domains of the FSFI measure and refer to the *most recent* sexual encounter.

[3]Correlations were not computed for sexual desire, arousal, or satisfaction because they pertain to an ideal sexual encounter, not the most recent sexual encounter.

during the most recent sexual encounter, relationship commitment, and relationship satisfaction. Results indicated that partner mate value ($B = .45$, $SE = .13$, $p = .001$); self mate value ($B = .86$, $SE = .12$, $p < .001$); commitment ($B = .54$, $SE = .21$, $p = .01$); and relationship satisfaction ($B = .36$, $SE = .11$, $p < .001$) were all independent, significant predictors for sexiness of underclothes. However, when controlling for partner mate value, commitment, and relationship satisfaction, self mate value was the only predictor for sexiness of underclothes ($B = .87$, $SE = .14$, $p < .001$), indicating a full mediating effect of self mate value on intimate apparel use in context two. All these results show that a woman's own mate value predicts her use of intimate apparel during her most recent sexual encounter despite her partner's mate value, her commitment to the relationship, and her relationship satisfaction.

Further analysis showed similar results for both contexts one and three, indicating that a woman's perceived mate value determines her use of intimate apparel outside the context of her current romantic relationship. That said, an analysis of relationship commitment indicates that women who are more committed to their relationship *and* have a negative MVD (their

**Table 5. Means and standard deviations for reported reasons for intimate apparel.**

| *Strongly disagree–Strongly agree* | n | M | SD | Min | Max |
|---|---|---|---|---|---|
| **I wear sexy underclothes:** | | | | | |
| to feel sexy. | 313 | **5.65** | 1.32 | 1 | 7 |
| because my partner tells me to. | 313 | 2.89 | 1.83 | 1 | 7 |
| on special occasions. | 313 | **5.41** | 1.39 | 1 | 7 |
| to feel feminine. | 313 | **5.28** | 1.52 | 1 | 7 |
| because that's what women are supposed to do for their partners | 313 | 3.14 | 1.94 | 1 | 7 |
| because it's comfortable. | 313 | 3.79 | 1.74 | 1 | 7 |
| to sleep in. | 313 | 3.44 | 1.97 | 1 | 7 |
| underneath my normal, everyday clothes. | 312 | 3.73 | 1.96 | 1 | 7 |
| when I expect to get intimate later that day. | 313 | **5.20** | 1.58 | 1 | 7 |
| as a gift for my partner. | 313 | **5.24** | 1.71 | 1 | 7 |
| when I run out of normal, everyday underwear. | 313 | 3.41 | 1.92 | 1 | 7 |
| underneath special, sexy clothes. | 313 | **5.01** | 1.79 | 1 | 7 |

Means are bolded if they are greater than four, meaning respondents agreed with the statements.

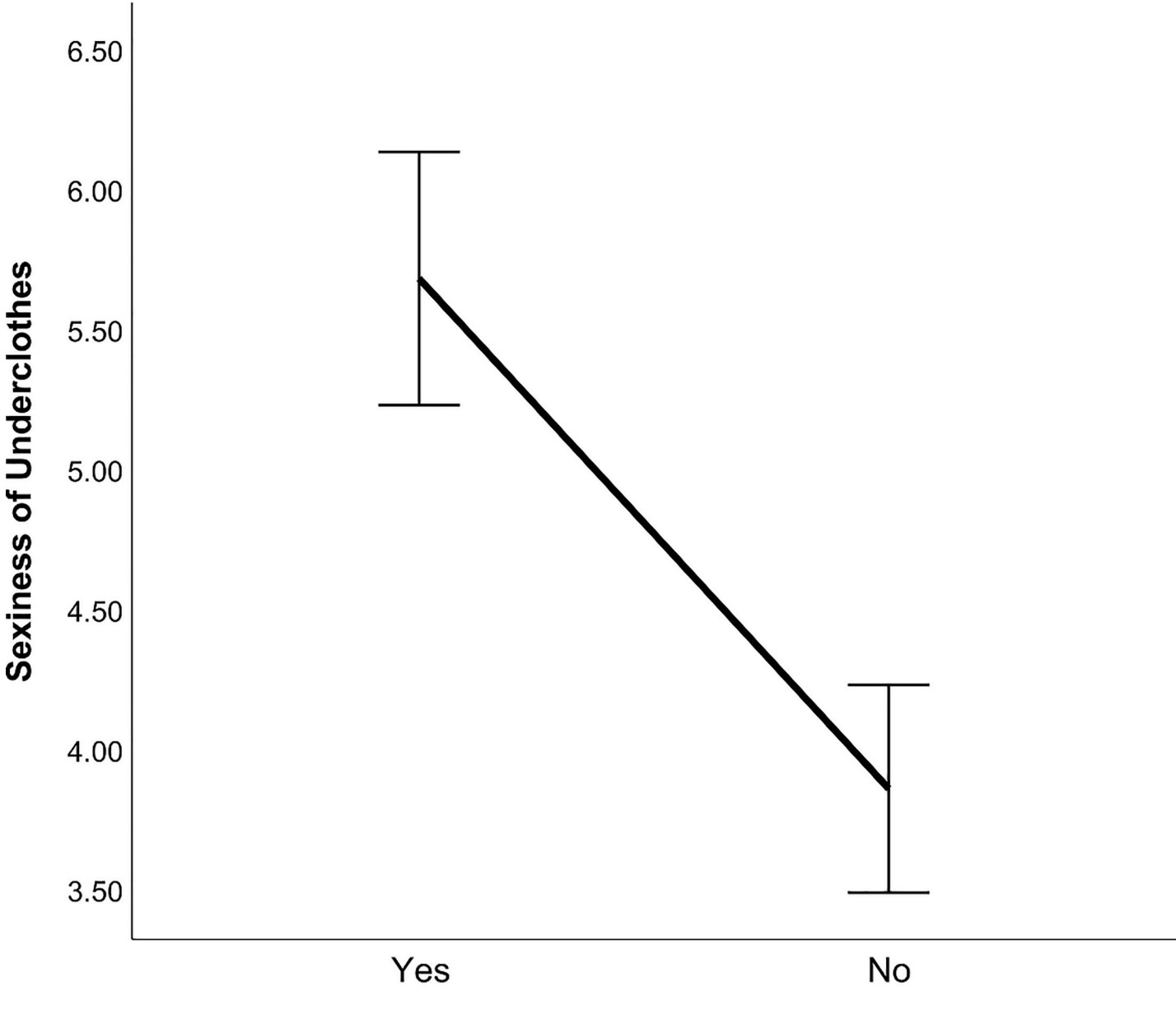

**Fig 1. Expect to be sexually active and score for sexiness of underclothes.** Error bars are at 95% confidence interval. The y-axis represents the mean scores for sexiness of underclothes in Context One.

own mate value is less than their partner's) engaged in more use of sexy underclothes during their most recent sexual encounter ($B$ = -.35, $SE$ = .16, $p$ = .03) but not during contexts one or three.

## Discussion

The purpose of this study was to identify women's use of intimate apparel as a subtle sexual signal of proceptivity and receptivity in heterosexual romantic relationships; however, our

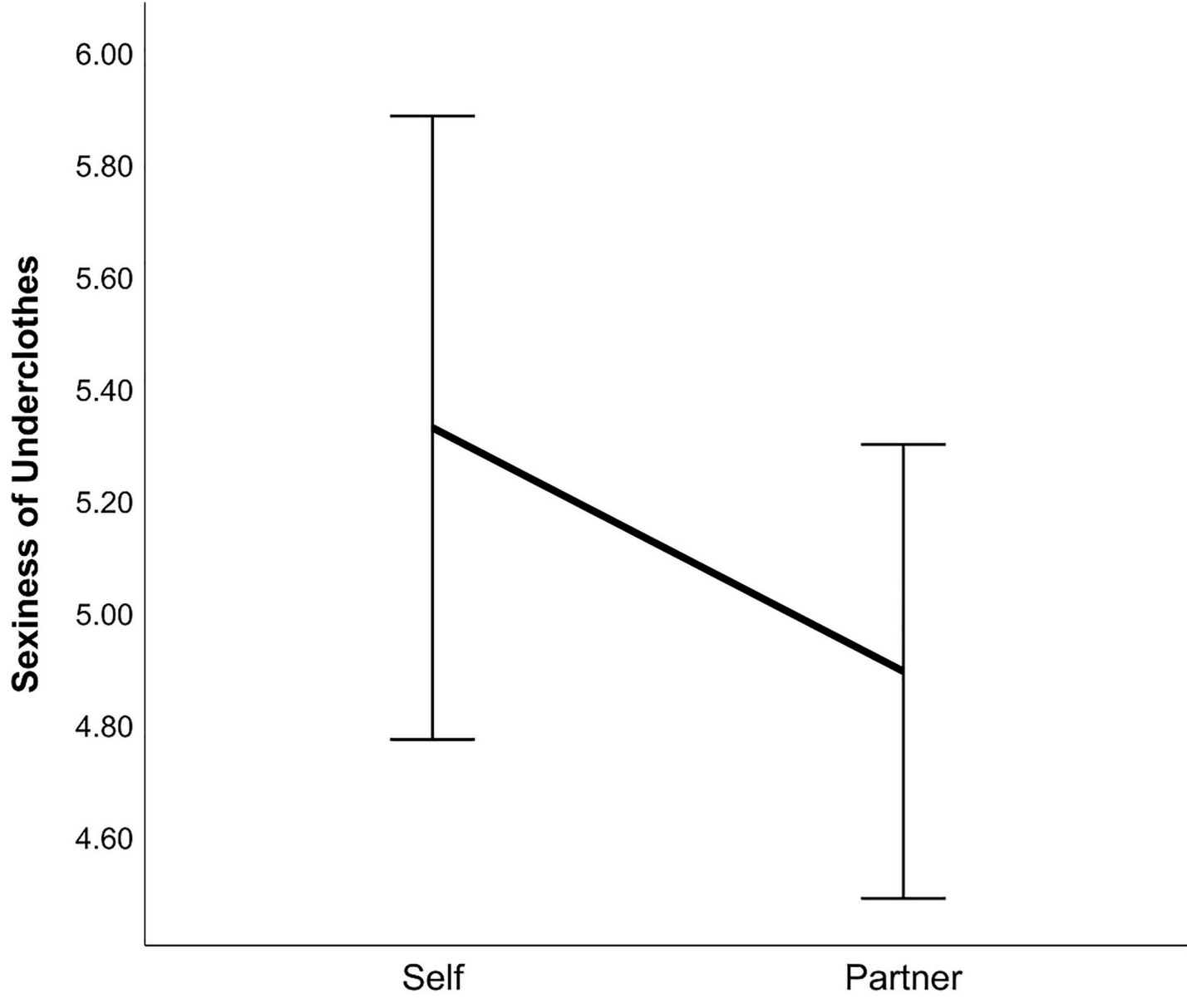

**Fig 2. Initiation of self or partner and score for sexiness of underclothes.** Error bars are at 95% confidence interval. The y-axis represents the mean scores for sexiness of underclothes in Context Two.

findings suggest intimate apparel might serve another purpose for the women in this study. We predicted that women would wear sexier underwear when they expected to be sexually active and when initiating sexual activity. Women in this study did indeed report wearing sexier underwear when intending to be sexually active later that day, but women's use of intimate apparel did not significantly contribute to sexual initiation. We also predicted that if a woman's perceived mate value was relatively lower than her partner's, she would be more likely to use intimate apparel to signal her proceptivity and receptivity. Conversely, we found that when

a woman's relative mate value was similar to or higher than–but not less than–her partner's mate value, she reported wearing sexier underwear across all three contexts.

Further analyses into MVD found that women in relationships with men of higher mate value than themselves, and relatively high independent mate values, felt more overall satisfied and committed to the relationship; but her partner's relative mate value did not influence her sexual experiences. Additionally, a woman's independent mate value predicted the sexiness of her underwear despite her commitment, satisfaction, and perception of her partner's mate value. Based on relative mate value, women might use intimate apparel when they feel more confident about themselves and their sexuality rather than as a signal of proceptivity and receptivity to their partner. So, why do women use intimate apparel if not for sexual signaling?

The number one reason women in this study reported wearing sexy underwear was "to feel sexy," which is consistent with other findings on intimate apparel [42, 43]. Use of intimate apparel might also act as a form of self-arousal: a thorough review of women's sexual desire by Meana [55] suggests that women want to feel sexually desired. Although a main factor in the decline of sexual desire and overall functioning for women is the presence of children and increased roles and responsibilities [56–58], the current study found that women's parental status, employment status, income, and age were not predictors for the use of intimate apparel. We found the best predictors for women's use of intimate apparel during their most sexual encounter to be relative mate value, relationship commitment, and relationship satisfaction. Other reasons reported for wearing intimate apparel were as a gift for their partner ($M = 5.24$, $SD = 1.71$) or on special occasions ($M = 5.41$, $SD = 1.39$). Furthermore, Pearson correlation analysis showed women who wore sexier underwear reported greater sexual desire ($r = .218$, $p < .001$) within the last four weeks and greater sexual desire ($r = .271$, $p < .001$), arousal ($r = .291$, $p < .001$), and satisfaction ($r = .204$, $p < .001$) during their most recent sexual encounter. Because there is a decline in sexual desire with relationship duration [57, 59], these women might be using intimate apparel to increase their own sexual desire, thus enhancing sexual encounters with their romantic partner [60] and maintaining the relationship they are committed to. Taken together, these results suggest women might use intimate apparel to increase their attractivity, which reflects a background signal of availability to romantic partners.

## Limitations and future research

This study is subject to limitations. These data worked under the assumption that the respondents had only one romantic partner; extra pair partners could potentially influence how women use intimate apparel in their daily lives as well as within their committed relationships. Results were also not adjusted for time-of-day or cycle phase. A methodological limitation to this study is that participants were not asked the duration of their relationship, so use of intimate apparel across time could not be measured. Schmiedeberg and Shroder [61] found that sexual satisfaction peaks in the second half of the first year of a relationship and then declines over time. As the relationship progresses, the couple becomes more acquainted with one another and experience mutual life stressors; the couple transitions into the companionate phase, characterized by a state of comfort and commonality [62]. However, Frederick, Lever, Gillespie and Garcia [60] found that passion in romantic relationships can be maintained by enhancing the quality of the sexual encounters with sensual touch and "I love you" statements, so use of intimate apparel might also serve to maintain passion.

Another methodological limitation was the measure for initiation (mentioned in the Results), which was one item asking, "Did you or your partner initiate your most recent sexual encounter?" without a clear definition of *initiate*. A more specific measure of MVD beyond self and partner desirability would provide greater insight into individuals' perceptions of

themselves and their partners. Because this study focused on heterosexual women, future studies should examine how lesbian women use intimate apparel in their short- and long-term romantic relationships. Finally, this study does not provide support for intimate apparel as a form of proceptivity; future research might ask more probing question about sexual activities, specifically in open-ended interviews.

These findings might also have clinical significance in that women's use of intimate apparel could reflect women's body image. In a comprehensive review on women's body image and sexual functioning, Woetman and ban den Brink [58] found a positive relationship between body image and sexual desire overall. They also found that women's sexual arousal and satisfaction can be negatively affected by self-inspection and -evaluation during sexual activity. Another study on body image and romantic relationships found that body appreciation was positively associated with relationship quality as well as sexual satisfaction [63]. Because women in this study who reported higher mate value were more likely to wear sexier underwear, body image might play a role in women's choice of intimate apparel and overall impact relationship and sexual satisfaction. Women who have lower body image might not have the confidence to wear sexier underwear, which could itself negatively affect her sexual desire and arousal. Future research into effects of body image should incorporate women's use of intimate apparel as well as mate value to better understand these associations.

## Conclusions

Little research has examined women's use of intimate apparel, and no studies have addressed its use as a subtle sexual signal in committed, romantic relationships. Despite some claims that use of intimate apparel is the result of effective marketing [33], we have found that the women in this study want to wear sexy underclothes to feel sexy, to feel desired, to feel aroused, and to prepare for sex with their partners. We have found that mate value has an important, albeit surprising, role in relationships regarding intimate apparel. Overall, a woman's use of intimate apparel is more closely related to her perceptions of her own mate value than her committed relationship. However, she still might use intimate apparel as a form of relationship maintenance: intimate apparel increases a woman's attractivity, which may function as an underlying signal of availability and receptivity. Although this study failed to establish proceptivity, it is the first to examine women's use of intimate apparel within romantic relationships.

## Acknowledgments

We would like to acknowledge the important theoretical contributions provided by LKC's graduate committee from the University of Nevada, Las Vegas (UNLV): Dr. Alyssa Crittenden, Dr. Jiemin Bao, and Dr. Murray Millar. For his substantial assistance with the initial power analysis, we would also like to thank Dr. Chad Cross from the UNLV School of Medicine.

## Author Contributions

**Conceptualization:** Lyndsey K. Craig, Peter B. Gray.

**Data curation:** Lyndsey K. Craig.

**Formal analysis:** Lyndsey K. Craig, Peter B. Gray.

**Funding acquisition:** Lyndsey K. Craig.

**Investigation:** Lyndsey K. Craig.

**Methodology:** Lyndsey K. Craig, Peter B. Gray.

**Project administration:** Lyndsey K. Craig.

**Resources:** Lyndsey K. Craig.

**Software:** Lyndsey K. Craig.

**Supervision:** Peter B. Gray.

**Validation:** Peter B. Gray.

**Visualization:** Lyndsey K. Craig.

**Writing – original draft:** Lyndsey K. Craig.

**Writing – review & editing:** Lyndsey K. Craig, Peter B. Gray.

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
