## [Decision Letter · Decision Letter 0]

16 Dec 2019

PONE-D-19-31911

Woman’s use of intimate apparel as subtle sexual signals in committed, heterosexual relationships

PLOS ONE

Dear Ms. Craig,

Thank you for submitting your manuscript to PLOS ONE. After careful consideration, we feel that it has merit but does not fully meet PLOS ONE’s publication criteria as it currently stands. Therefore, we invite you to submit a revised version of the manuscript that addresses the points raised during the review process.

We would appreciate receiving your revised manuscript by Jan 30 2020 11:59PM. To enhance the reproducibility of your results, we recommend that if applicable you deposit your laboratory protocols in protocols.io, where a protocol can be assigned its own identifier (DOI) such that it can be cited independently in the future. For instructions see: http://journals.plos.org/plosone/s/submission-guidelines#loc-laboratory-protocols

We look forward to receiving your revised manuscript.

Kind regards,

Amir H. Pakpour, Ph.D.

Academic Editor

PLOS ONE

Additional Editor Comments (if provided):

While the topic is interesting but it's clinical implication is not well defined. The authors need to clarify this in the manuscript throughout. 

Journal Requirements:

Reviewers' comments:

Reviewer's Responses to Questions

**Comments to the Author**

1. Is the manuscript technically sound, and do the data support the conclusions?

Reviewer #1: Yes

2. Has the statistical analysis been performed appropriately and rigorously? 

Reviewer #1: Yes

3. Have the authors made all data underlying the findings in their manuscript fully available?

Reviewer #1: Yes

4. Is the manuscript presented in an intelligible fashion and written in standard English?

Reviewer #1: Yes

5. Review Comments to the Author

Reviewer #1: Introduction: Explanation regarding the prices (line 133-138) does not seem to be necessary.

Methods: psychometric properties of measures is not explained.linear regression has some assumptions. a brief explanation of how these assumptions were checked on this data might be useful for readers.

Discussion: In this section main findings are explained but in some parts the findings are repeated. referring to tables is not appropriate for this part. the discussion should be enriched by adding more comparison with previous literature. It is seems better to add some explanation regarding the clinical implication of present findings.

6. PLOS authors have the option to publish the peer review history of their article (what does this mean?). If published, this will include your full peer review and any attached files.

Reviewer #1: No

---

## [Author Response · Author response to Decision Letter 0]

27 Jan 2020

Editor Comments: 

“While the topic is interesting but it’s clinical implication is not well defined. The authors need to clarify this in the manuscript throughout.”

We thank the editor for their interest in this topic. To address the issue of clinical implication, we have included a paragraph under the Limitations and Future Research section (lines 539-551) to illustrate possible clinical significance of women’s use of intimate apparel.

“Please ensure that your manuscript meets PLOS ONE’s style requirements, including those for file naming.”

Per the editor’s comment, we have gone through the entire document to ensure it meets all style requirements of PLOS ONE including minor edits to the Title Page, elimination of Keywords section, and complete revision of tables and figures. We have also run the figures through the PACE digital diagnostic tool after reconfiguration. To the best of our knowledge, the manuscript now meets all style requirements. 

Reviewer Comments:

“Introduction: Explanation regarding the prices (line 133-138) does not seem to be necessary.”

The reviewer’s comment is well-taken, though we still believe it important to emphasize the substantial cost of intimate apparel. Therefore, we removed lines 133-138 and added a simpler phrase to the sentence on line 131: “Although a more than 30 billion U.S. dollar market worldwide…”

“Methods: psychometric properties of measures is not explained. linear regression has some assumptions. a brief explanation of how these assumptions were checked on this data might be useful for readers.”

We agree with this comment and have added notes of reliability and validity to all relevant measures and made the methods section more transparent, such as separating the Intimate Apparel section to two sections: Attitudes and Use. We also created a separate section for the three Contexts. In the Statistical Analysis section, we specified how we measured intent and initiation (line 306) and provided a brief explanation about how we checked the assumptions of normality against using our statistical tests (lines 307-309).

“Discussion: In this section main findings are explained but in some parts the findings are repeated. referring to tables is not appropriate for this part. the discussion should be enriched by adding more comparison with previous literature. It is seems better to add some explanation regarding the clinical implication of present findings.”

This comment highlights flaws in the construction of the Discussion section. So, we have completely revised the Discussion section to eliminate redundancy and provide a logical flow of information. We have also removed any references to tables and have better incorporated previous literature.

As we mentioned in the editor comments, to address the issue of clinical implication, we have included a paragraph under the Limitations and Future Research section (lines 539-551) to illustrate possible clinical significance of women’s use of intimate apparel.

---

## [Decision Letter · Decision Letter 1]

24 Feb 2020

Women's use of intimate apparel as subtle sexual signals in committed, heterosexual relationships

PONE-D-19-31911R1

Dear Dr. Craig,

We are pleased to inform you that your manuscript has been judged scientifically suitable for publication and will be formally accepted for publication once it complies with all outstanding technical requirements.

With kind regards,

Amir H. Pakpour, Ph.D.

Academic Editor

PLOS ONE

Additional Editor Comments (optional):

Reviewers' comments:

Reviewer's Responses to Questions

**Comments to the Author**

1. If the authors have adequately addressed your comments raised in a previous round of review and you feel that this manuscript is now acceptable for publication, you may indicate that here to bypass the “Comments to the Author” section, enter your conflict of interest statement in the “Confidential to Editor” section, and submit your "Accept" recommendation.

Reviewer #1: (No Response)

Reviewer #2: All comments have been addressed

2. Is the manuscript technically sound, and do the data support the conclusions?

Reviewer #1: (No Response)

Reviewer #2: Yes

3. Has the statistical analysis been performed appropriately and rigorously? 

Reviewer #1: (No Response)

Reviewer #2: Yes

4. Have the authors made all data underlying the findings in their manuscript fully available?

Reviewer #1: (No Response)

Reviewer #2: Yes

5. Is the manuscript presented in an intelligible fashion and written in standard English?

Reviewer #1: (No Response)

Reviewer #2: Yes

6. Review Comments to the Author

Reviewer #1: (No Response)

Reviewer #2: Although I have not reviewed the original submission of the manuscript, I found that the authors have well addressed the comments made from the previous reviewer. Therefore, I feel that the current resubmission has high quality and should be fulfilled the publication criteria.

7. PLOS authors have the option to publish the peer review history of their article (what does this mean?). If published, this will include your full peer review and any attached files.

Reviewer #1: No

Reviewer #2: No

---

## [Editor Report · Acceptance letter]

3 Mar 2020

PONE-D-19-31911R1 

Women's use of intimate apparel as subtle sexual signals in committed, heterosexual relationships 

Dear Dr. Craig:

I am pleased to inform you that your manuscript has been deemed suitable for publication in PLOS ONE. Congratulations! Your manuscript is now with our production department. 

With kind regards,

on behalf of

Dr. Amir H. Pakpour 

Academic Editor

PLOS ONE